# Associations between insomnia and pregnancy and perinatal outcomes: Evidence from mendelian randomization and multivariable regression analyses

Qian Yang[1,2]*, Maria Carolina Borges[1,2], Eleanor Sanderson[1,2], Maria C. Magnus[1,2,3], Fanny Kilpi[1,2], Paul J. Collings[4], Ana Luiza Soares[1,2], Jane West[4], Per Magnus[3], John Wright[4], Siri E. Håberg[3], Kate Tilling[1,2,5], Deborah A. Lawlor[1,2,5]

**1** MRC Integrative Epidemiology Unit at the University of Bristol, Bristol, United Kingdom, **2** Population Health Sciences, Bristol Medical School, University of Bristol, United Kingdom, **3** Centre for Fertility and Health, Norwegian Institute of Public Health, Oslo, Norway, **4** Bradford Institute for Health Research, Bradford Teaching Hospitals NHS Foundation Trust, Bradford, United Kingdom, **5** National Institute for Health Research Bristol Biomedical Centre, University Hospitals Bristol NHS Foundation Trust and University of Bristol, Bristol, United Kingdom

* qian.yang@bristol.ac.uk

**Data Availability Statement:** We used both individual participant cohort data and publicly

## Abstract

### Background

Insomnia is common and associated with adverse pregnancy and perinatal outcomes in observational studies. However, those associations could be vulnerable to residual confounding or reverse causality. Our aim was to estimate the association of insomnia with stillbirth, miscarriage, gestational diabetes (GD), hypertensive disorders of pregnancy (HDP), perinatal depression, preterm birth (PTB), and low/high offspring birthweight (LBW/HBW).

### Methods and findings

We used 2-sample mendelian randomization (MR) with 81 single-nucleotide polymorphisms (SNPs) instrumenting for a lifelong predisposition to insomnia. Our outcomes included ever experiencing stillbirth, ever experiencing miscarriage, GD, HDP, perinatal depression, PTB (gestational age <37 completed weeks), LBW (<2,500 grams), and HBW (>4,500 grams). We used data from women of European descent ($N = 356,069$, mean ages at delivery 25.5 to 30.0 years) from UK Biobank (UKB), FinnGen, Avon Longitudinal Study of Parents and Children (ALSPAC), Born in Bradford (BiB), and the Norwegian Mother, Father and Child Cohort (MoBa). Main MR analyses used inverse variance weighting (IVW), with weighted median and MR-Egger as sensitivity analyses. We compared MR estimates with multivariable regression of insomnia in pregnancy on outcomes in ALSPAC ($N = 11,745$). IVW showed evidence of an association of genetic susceptibility to insomnia with miscarriage (odds ratio (OR): 1.60, 95% confidence interval (CI): 1.18, 2.17, $p = 0.002$), perinatal depression (OR 3.56, 95% CI: 1.49, 8.54, $p = 0.004$), and LBW (OR 3.17, 95% CI: 1.69, 5.96, $p < 0.001$). IVW results did not support associations of insomnia with stillbirth, GD, HDP, PTB,

available summary statistics. We present summary statistics that we generated from those individual participant cohort data in S4 and S5 Tables. Full information on how to access UKB data can be found at its website (https://www.ukbiobank.ac.uk/researchers/). All ALSPAC data are available to scientists on request to the ALSPAC Executive via this website (http://www.bristol.ac.uk/alspac/researchers/), which also provides full details and distributions of the ALSPAC study variables. Similarly, data from BiB are available on request to the BiB Executive (https://borninbradford.nhs.uk/research/how-to-access-data/). Data from MoBa are available from the Norwegian Institute of Public Health after application to the MoBa Scientific Management Group (see its website https://www.fhi.no/en/op/data-access-from-health-registries-health-studies-and-biobanks/data-access/applying-for-access-to-data/ for details). Summary statistics from FinnGen are publicly available on its website (https://finngen.gitbook.io/documentation/data-download).

**Funding:** QY, MCB, ES, FK, ALS, KT and DAL work in a unit that is supported by the University of Bristol and UK Medical Research Council (MRC, MM_UU_00011/1, MM_UU_00011/3 to KT and MM_UU_00011/6 to DAL). This work was supported by China Scholarship Council PhD Scholarship (CSC201807060273 to QY), UK MRC Skilled Development Fellowship (MR/P014054/1 to MCB), Research Council of Norway through its Centres of Excellence funding scheme (262700 to MCM and SHE), European Research Council (ERC) under the European Union's Horizon 2020 research and innovation programme (947684 to MCM), and British Heart Foundation (BHF) Immediate Postdoctoral Basic Science Research Fellowship (FS/17/37/32937 to PJC). DAL is a BHF Chair (CH/F/20/9003) and National Institute of Health Research (NIHR) Senior Investigator (NF-0616-10102). Core funding for ALSPAC is provided by UK MRC and University of Bristol (217065/Z/19/Z). Genotyping of maternal samples was funded by the Wellcome Trust (WT088806), and offspring samples were genotyped by Sample Logistics and Genotyping Facilities at the Wellcome Trust Sanger Institute and LabCorp (Laboratory Corporation of America) using support from 23andMe. A comprehensive list of grants funding is available on the ALSPAC website (http://www.bristol.ac.uk/alspac/external/documents/grant-acknowledgements.pdf). BiB is supported by a Wellcome programme grant (WT223601/Z/21/Z: Age of Wonder), an infrastructure grant (WT101597MA), a UK MRC and Economic and Social Science Research Council programme grant (MR/N024397/1), a BHF Clinical Study grant (CS/

and HBW, with wide CIs including the null. Associations of genetic susceptibility to insomnia with miscarriage, perinatal depression, and LBW were not observed in weighted median or MR-Egger analyses. Results from these sensitivity analyses were directionally consistent with IVW results for all outcomes, with the exception of GD, perinatal depression, and PTB in MR-Egger. Multivariable regression showed associations of insomnia at 18 weeks of gestation with perinatal depression (OR 2.96, 95% CI: 2.42, 3.63, $p < 0.001$), but not with LBW (OR 0.92, 95% CI: 0.69, 1.24, $p = 0.60$). Multivariable regression with miscarriage and stillbirth was not possible due to small numbers in index pregnancies. Key limitations are potential horizontal pleiotropy (particularly for perinatal depression) and low statistical power in MR, and residual confounding in multivariable regression.

## Conclusions

In this study, we observed some evidence in support of a possible causal relationship between genetically predicted insomnia and miscarriage, perinatal depression, and LBW. Our study also found observational evidence in support of an association between insomnia in pregnancy and perinatal depression, with no clear multivariable evidence of an association with LBW. Our findings highlight the importance of healthy sleep in women of reproductive age, though replication in larger studies, including with genetic instruments specific to insomnia in pregnancy are important.

## Author summary

### Why was this study done?

- Insomnia in pregnancy was associated with higher risks of adverse pregnancy and perinatal outcomes in observational studies.

- It is currently not clear whether insomnia causes adverse pregnancy and perinatal outcomes or whether the unfavourable associations are explained by confounding.

- To the best of our knowledge, mendelian randomization (MR) has not been used to explore whether there is evidence to support a causal association between insomnia and adverse pregnancy and perinatal outcomes.

### What did the researchers do and find?

- We used data on up to 356,069 women from UK Biobank (UKB), FinnGen, and 3 birth cohorts and assessed whether genetic susceptibility to insomnia was associated with stillbirth, miscarriage, gestational diabetes (GD), hypertensive disorders of pregnancy (HDP), perinatal depression, preterm birth (PTB), low offspring birthweight (LBW), and high offspring birthweight (HBW) in 2-sample MR.

- To triangulate with our MR estimates, we conducted multivariable regression in 11,745 women from the Avon Longitudinal Study of Parents and Children (ALSPAC), where

16/4/32482), and NIHR under its Applied Health Research Collaboration Yorkshire and Humber (NIHR200166) and the NIHR Clinical Research Network. Further supports for genome-wide and multiple omics measurements in BiB are from UK MRC (G0600705), NIHR (NF-SI-0611010196), US National Institute of Health (R01DK10324), and ERC via Advanced Grant (669545) and under the European Union's Seventh Framework Programme (FP7/2007-2013). The funders had no role in study design, data collection and analysis, decision to publish or preparation of the manuscript.

**Competing interests:** I have read the journal's policy and the authors of this manuscript have the following competing interests: KT has acted as a consultant for CHDI Foundation, and Expert Witness to the High Court in England, called by the UK Medicines and Healthcare products Regulatory Agency, defendants in a case on hormonal pregnancy tests and congenital anomalies 2021/22. DAL has received support from Medtronic LTD and Roche Diagnostics for biomarker research that is not related to the study presented in this paper. The other authors report no conflicts.

**Abbreviations:** ALSPAC, Avon Longitudinal Study of Parents and Children; BiB, Born in Bradford; CI, confidence interval; GD, gestational diabetes; GWAS, genome-wide association study; HBW, high offspring birthweight; HDP, hypertensive disorders of pregnancy; IV, instrumental variable; IVW, inverse variance weighting; LBW, low offspring birthweight; MoBa, the Norwegian Mother, Father and Child Cohort; MR, mendelian randomization; OR, odds ratio; PC, principal component; PTB, preterm birth; SNP, single-nucleotide polymorphism; STROBE, Strengthening the Reporting of Observational Studies in Epidemiology; UKB, UK Biobank.

insomnia was measured in pregnancy for all outcomes except miscarriage and stillbirth for which there were too few cases in the index pregnancy.

- We found evidence from MR and multivariable regression that insomnia was associated with a higher risk of perinatal depression, and MR analyses also suggested evidence for an association between genetically predicted insomnia and risks of miscarriage and LBW.

## What do these findings mean?

- These findings raise the possibility that insomnia maybe related to adverse pregnancy outcomes, implying that interventions to improve healthy sleep may be beneficial to a healthy pregnancy.

- Key limitations of our study are potential horizontal pleiotropy (particularly for perinatal depression) and low statistical power in MR and residual confounding in multivariable regression. Replication in larger MR studies would be valuable.

## Introduction

Insomnia, which affects approximately 10% to 20% of the adult population, is usually defined as a difficulty in getting to sleep or remaining asleep, or having a nonrestorative sleep, and such sleep impairment can be associated with daytime sleepiness [1,2]. Physical and hormonal changes during pregnancy increase susceptibility to insomnia [3,4].

Most evidence on the relationship between insomnia during pregnancy and adverse pregnancy and perinatal outcomes has come from observational studies. The most recently updated systematic reviews of observational studies suggest that pregnancy-related insomnia and poor sleep quality are associated with higher risks of gestational diabetes (GD) [5,6], hypertensive disorders of pregnancy (HDP) [6], perinatal depression [7], and preterm birth (PTB) [6]. Other observational studies have shown that specific conditions that relate to insomnia are also associated with adverse pregnancy and perinatal outcomes. Sleep-disordered breathing, obstructive sleep apnoea, and restless legs syndrome have also been shown to associate with higher risks of GD, HDP, large-for-gestational age, and low offspring birthweight (LBW) [6]. However, it remains unclear whether insomnia causes adverse pregnancy outcomes or whether these associations are explained by confounding, e.g., due to socioeconomic status and lifestyle factors. It is also possible that some of these studies reflect reverse causation. For example, all 4 studies included in the systematic review for perinatal depression were cross-sectional [7], in which disturbed sleep could be either a symptom of or a risk factor for depression. Furthermore, most individual studies focus on just 1 or 2 outcomes. Examining potential effects on a range of adverse pregnancy and perinatal outcomes is important to understand the overall health impact of insomnia during pregnancy.

Three randomized control trials assessing the effects of interventions to prevent insomnia on adverse pregnancy and perinatal outcomes have been published [8–10]. All 3 of these used cognitive behavioural interventions targeted at reducing insomnia, with the primary outcome being Edinburgh Postnatal Depression Scale scores. The small number of randomized control

trials, their small sample sizes, and directional inconsistency, but overlapping 95% confidence intervals (CIs), make it difficult to draw conclusions, and none of them explored other adverse pregnancy or perinatal outcomes.

Mendelian randomization (MR) provides an alternative way to assess the impact of insomnia on adverse pregnancy and perinatal outcomes by using genetic variants (mostly single-nucleotide polymorphisms [SNPs]) as instrumental variables (IVs) for insomnia [11,12]. MR is less prone to confounding than observational studies, as genetic variants are randomly allocated at meiosis and cannot be influenced by the wide range of sociodemographic or behavioural factors which conventionally confound observational studies nor can they be influenced by health status [11,12]. Under key assumptions (discussed in Methods), MR can be used to estimate a causal association from the SNPs-exposure and SNPs-outcome associations, if the underlying assumptions (in Discussion) are true. In 2-sample MR, the SNP-exposure and SNP-outcome associations are estimated using different (ideally independent) studies from the same underlying population [13]. This approach has previously been used to evaluate causal associations of insomnia with type 2 diabetes [14,15], hypertension [16], and cardiovascular disease [15,17,18] in non-pregnant populations, but to the best of our knowledge not pregnancy and perinatal outcomes.

The aims of this study are to (I) explore the causal associations of maternal genetic susceptibility to insomnia with stillbirth, miscarriage, GD, HDP, perinatal depression, PTB, LBW, and high offspring birthweight (HBW), using 2-sample MR; and (II) compare MR findings with conventional multivariable regression analyses of self-reported insomnia during pregnancy with these outcomes, where possible.

## Methods

### Study populations

This study was undertaken using data from the MR-PREG collaboration, which aims to explore causes and consequences of different pregnancy and perinatal outcomes [19]. We used individual-level data from UK Biobank (UKB) women ($N$ = 208,140, recruited between 2006 to 2010) and mother-offspring pairs from Avon Longitudinal Study of Parents and Children (ALSPAC, $N$ = 6,826, recruited between 1991 to 1992), Born in Bradford (BiB, $N$ = 2,940, recruited between 2007 to 2010), and the Norwegian Mother, Father and Child Cohort (MoBa, $N$ = 14,584, recruited between 1999 to 2009). To be comparable across all cohorts, only genetically unrelated women of European descent with qualified genotype data (and with singleton offspring in birth cohorts) were eligible for inclusion in our analyses (S1 Fig). We also used summary-level genetic association data from FinnGen—the national wide network of Finnish biobanks ($N$ = up to 123,579 women) [20]. All studies had ethical approval from relevant national or local bodies and participants provided written informed consent. Details of the recruitment, information on genetic data, and measurements of baseline characteristics of each cohort are described in S1 Text. This study was initiated using UKB in January 2018, with extra exploration of insomnia IVs and MR sensitivity analyses completed in February 2020 [21]. We searched for additional cohorts till July 2021, and harmonization across the cohorts had to be made continuously. Therefore, we did not have a prespecified analysis plan.

### Outcomes measures

We explored potential effects of insomnia on 8 binary outcomes: ever experiencing stillbirth, ever experiencing miscarriage, GD, HDP, perinatal depression, PTB (gestational age <37 completed weeks), LBW (<2,500 grams), and HBW (>4,500 grams). Full details about how these outcomes were measured and derived in each participating study and how we harmonised

them across studies can be found in S1 Table. We were not able to measure pre-eclampsia and gestational hypertension separately, because of the small number of definite cases of pre-eclampsia, and because of differences between studies in data collection and definitions.

In UKB, gestational age was only available for a small subset of women (N = 7,280) who delivered a child during or after 1989, the earliest date for which linked hospital labour and perinatal data are available [22]. As a result, numbers with data on PTB are smaller than for any other outcome, and we a priori decided to examine associations with LBW and HBW rather than small-for-gestational age and large-for-gestational age. For most outcomes in UKB, women reported their experience retrospectively in a questionnaire completed at recruitment when they were aged 40 to 60 years.

In the 3 birth cohorts, most outcomes were prospectively obtained (from self-report or clinical records) during an index pregnancy and the perinatal period. The 2 exceptions were history of stillbirth and miscarriage, which were retrospectively reported at the time of the index pregnancy when women were asked if they had ever experienced a (previous) stillbirth or miscarriage. We explored the possibility of examining associations with miscarriage and stillbirth in the index pregnancy. However, numbers were too small for reliable results, and for miscarriage, we were concerned about misclassification or selection bias due to women who had experienced a miscarriage prior to recruitment. Therefore, we used MR to explore the association of susceptibility to insomnia on a history of miscarriage and stillbirth and did not undertake any multivariable regression analyses for these 2 outcomes as suggested during peer review. There were a small proportion of women who contributed more than 1 pregnancy (<5% of total N for each outcome). Given that choosing the first pregnancy could introduce selection towards younger age, lower parity, and higher morbidity of HDP [23], we followed EGG consortium convention [24] to randomly select 1 pregnancy per woman [25].

Data from FinnGen were available for 4 of our outcomes: ever experiencing miscarriage, GD, HDP, and PTB, which were defined based on International Classification of Diseases codes.

## Insomnia measures

Self-reported information on insomnia was obtained from 2 of the studies. In UKB, information on lifetime insomnia was used to generate SNP-insomnia associations in women for use in MR analyses in UKB and the birth cohorts. ALSPAC collected data on insomnia during pregnancy, and this was used for conventional confounder-adjusted multivariable regression.

In UKB, insomnia was self-reported at recruitment via the question "Do you have trouble falling asleep at night or do you wake up in the middle of the night?" with responses "never/rarely," "sometimes," "usually," and "prefer not to answer." For our analyses, we collapsed these categories to generate a binary variable of usually experiencing insomnia (i.e., "usually" [cases] versus "sometimes" + "never/rarely" [controls]) as this was how the responses were categorised in the published genome-wide association study (GWAS) that we have used to select genetic IVs [15].

In ALSPAC, insomnia in pregnancy was self-reported, at 18 and 32 weeks of gestation, using the question "Can you get off to sleep alright?" with options "Very often," "Often," "Not very often," and "Never." At each time point, we compared "Not very often" + "Never" [cases] versus "Very often" + "Often" [controls]. We acknowledge that the 2 studies are using different questions and that definitions of insomnia vary across published literature [2]. For ease of reading throughout the paper, we refer to results reflecting genetic susceptibility to insomnia (MR) and reporting insomnia in pregnancy (multivariable regression).

## SNP selection and SNP-insomnia associations

To identify genetic IVs for insomnia, we searched the GWAS published between January 2017 and February 2021 on PubMed and Neale Lab website [26]. We found 7 insomnia GWAS reporting genome-wide significant SNPs (details in S2 Table). Of these, we selected SNPs from the largest GWAS (total $N$ = 709,986 women, 29% from UKB, and 71% from 23andMe), which provided female-specific results [15]. This GWAS identified 83 loci containing 87 lead SNPs that were robustly associated with insomnia ($P$-value $< 5 \times 10^{-8}$) after pooling UKB and 23andMe women together. We removed 6 SNPs that were correlated to other SNPs (*linkage disequilibrium*) at an $R^2$ threshold of 0.01 or higher, based on all European samples from the 1,000 genome project [27]. Associations (reported in log odds ratios [ORs]) of the remaining 81 lead SNPs from the women only GWAS were extracted and listed in S3 Table.

We followed the standard IV approach [28], first using linear regression with individual-level data from 208,140 UKB women to obtain SNP-insomnia association summary data for 2-sample MR analyses. This provides estimates on the risk difference scale, which is more interpretable and comparable to our multivariable regression results [29]. We adjusted the linear models for genotyping batch, top 40 principal components (PCs) and women's age. During peer review, we were asked to regenerate SNP-insomnia associations using logistic regression to repeat MR analyses. Therefore, we reconducted: (I) split-sample analyses in UKB by generating SNP-insomnia and SNP-outcome associations in logistic regression; (II) 2-sample MR using SNP-insomnia associations generated in logistic regression by the GWAS where we selected our IVs [15], and the pooled SNP-outcome associations combining ALSPAC, BiB, MoBa, and FinnGen; and (III) a meta-analysis of MR estimates from (I) and (II) using fixed-effects (with inverse variance weights) for each insomnia-outcome pair. Consistent with a previous MR study of binary exposures [30], our MR estimates were reported as odds ratios per 1 unit higher log-odds of insomnia.

## SNP-outcome associations

We estimated the associations between maternal SNPs and outcomes (log OR and standard errors) for each of the 81 insomnia-related SNPs. In UKB, we randomly separating women in half (giving 2 datasets, A and B) for our split cross-over 2-sample MR [31], given UKB was also included in the GWAS of insomnia. We then estimated SNP-outcome associations in each split sample using logistic regression, adjusting for genotyping batch, top 40 PCs, and women's age. In the birth cohorts, we estimated the SNP-outcome associations using logistic regression, adjusting for (I) top 20 PCs and women's age in ALSPAC; (II) top 10 PCs and women's age in BiB; and (III) genotyping batch, top 10 PCs, and women's age in MoBa. We extracted associations of the 81 SNPs with the following from FinnGen (words in brackets are the outcome labels from FinnGen): miscarriage (O15_ABORT_SPONTAN), GD (GEST_DIABETES), HDP (O15_GESTAT_HYPERT), and PTB (O15_PRETERM). These summary data were generated by FinnGen using the R-package called SAIGE that fits mixed-effects logistic regression [32], adjusting for genotyping batch, top 10 PCs, and women's age [20]. Then, we meta-analysed associations from ALSPAC, BiB, MoBa, and FinnGen using fixed-effects with inverse variance weights. Two SNPs (i.e., rs10947428 and rs117037340) were excluded from BiB analyses due to their minor allele frequency lower than 1%.

## Assessment of confounders in ALSPAC for multivariable regression

We considered maternal age at time of delivery, education, body mass index at 12 weeks of gestation, smoking status in pregnancy, alcohol intake in the first 3 months of pregnancy, and

household occupational social class as potential confounders based on their known or plausible associations with maternal insomnia and pregnancy and perinatal outcomes. Details of confounders were based on maternal self-report and are fully described in S1 Text.

## Statistical analyses

**Two-sample MR.** As shown in Fig 1, we conducted 2-sample MR analyses of maternal insomnia on pregnancy and perinatal outcomes. In UKB, we conducted a split cross-over 2-sample MR [31]. Specifically, we used SNP-insomnia associations from dataset A and SNP-outcomes associations from dataset B (A on B) and vice-versa (B on A), and then meta-analysed the MR estimates from the 2 together for each insomnia-outcome pair using fixed-effects (with inverse variance weights). For the 2-sample MR using the rest of the cohorts, we used SNP-insomnia associations from UKB women and the pooled SNP-outcome associations combining ALSPAC, BiB, MoBa, and FinnGen. For each outcome, we pooled MR estimates from all cohorts using fixed-effects (with inverse variance weights) and used leave-one (study)-out analysis (initially across all cohorts and then among non-UKB cohorts during peer review) to assess the degree of heterogeneity between cohorts.

In the main analyses, we used the MR inverse variance weighting (IVW) method, which is a regression of the estimates for SNP-outcomes associations on SNP-insomnia associations weighted by the inverse of the SNP-outcome associations variances, with the intercept of the regression line forced through zero [33]. The IVW estimates should provide an unbiased

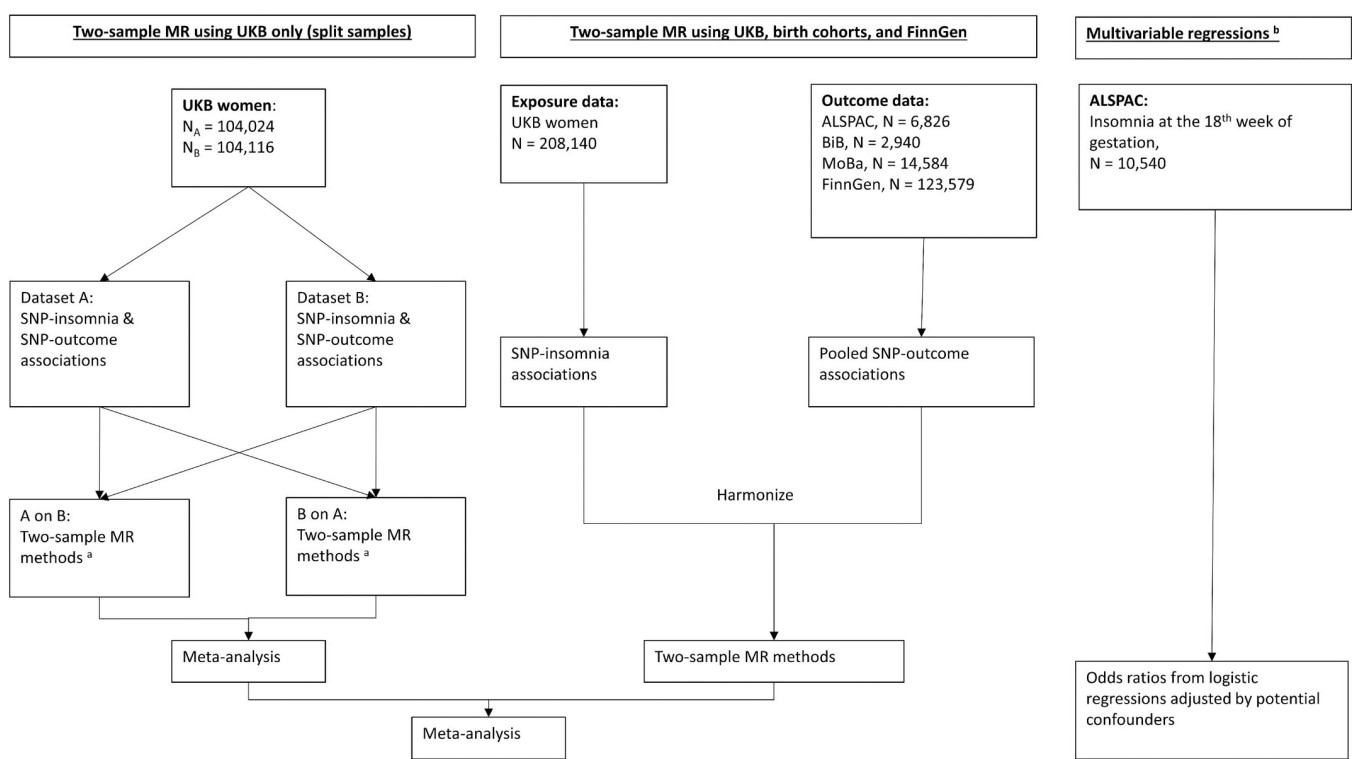

**Fig 1. Summary of methods and data contributing to this study.** (a) Two-sample MR methods include: IVW, MR-Egger, weighted median, and leave-one-out analysis. (b) Multivariable regression analysis adjusted for maternal age at time of delivery, social class, education, body mass index at 12 weeks of gestation, smoking status in pregnancy, and alcohol intake in the first 3 months of ALSPAC pregnancy. ALSPAC, Avon Longitudinal Study of Parents and Children; BiB, Born in Bradford; GWAS, genome-wide association study; IVW, inverse variance weighted; MoBa, Norwegian Mother, Father and Child Cohort Study; MR, mendelian randomization; SNP, single-nucleotide polymorphism; UKB, UK Biobank.

estimate of a causal effect in the absence of unbalanced horizontal pleiotropy [33]. To explore potential unbalanced horizontal pleiotropy, our sensitivity analyses included (I) estimating between-SNP heterogeneity (which if present may be due to one or more SNPs having horizontal pleiotropic effects on the outcome) using Cochran's Q-statistic and leave-one (SNP)-out analysis; and (II) undertaking analyses with weighted median [34] and MR-Egger [35], which are more likely to be robust in the presence of invalid IVs. The weighted median method is unbiased so long as less than 50% of the weight is from invalid instruments (i.e., if 1 SNP contributing more than 50% of the weight across the SNP-insomnia associations or several SNPs that contribute more than 50% introduce horizontal pleiotropy the effect estimate is likely to be biased) [34]. MR-Egger is similar to IVW except it does not constrain the regression line to go through zero; if the MR-Egger intercept is not null, it suggests the presence of unbalanced horizontal pleiotropy, and the MR-Egger slope provides an effect estimate corrected for unbalanced horizontal pleiotropy [35]. However, MR-Egger has considerably less statistical power than IVW. Further details of these MR methods are provided in our previous study [21]. When using MR to assess the effect of maternal exposures in pregnancy on offspring outcomes, results might be biased via a path from maternal genotypes to maternal/offspring outcomes due to fetal genotype [36]. To explore this, we compared SNP-outcome associations with versus without adjustments for fetal genotypes in the pooled birth cohort analyses.

We evaluated the strength of IVs using both proportion of variances of maternal insomnia explained by the 81 SNPs ($R^2$) and F-statistic [37]. We selected SNPs robustly related to insomnia in the general female population rather than in pregnant women. Therefore, we explored associations of the 81 SNPs with woman's insomnia measured at 18 and 32 weeks of gestation in ALSPAC using logistic regressions to determine whether those SNPs related similarly to insomnia in pregnancy. We adjusted for the top 20 PCs and women's age. As suggested during peer review, we used Steiger filtering to identify SNPs explaining substantially more of the variance in an outcome than in insomnia (i.e., $P$-value $< 0.05$) [38] and reconducted MR IVW after removing those SNPs (listed in S3 Table).

**Multivariable regression in ALSPAC.** In ALSPAC, we explored the observational associations of insomnia at 18 weeks of gestation with binary outcomes using logistic regression, with adjustment for measured confounders. During peer review, insomnia at 32 weeks of gestation was not considered in the analysis due to potential reverse causality for some outcomes.

All analyses were performed using R 3.5.1 (R Foundation for Statistical Computing, Vienna, Austria). Two-sample MR analyses were conducted using the "TwoSampleMR" R package [27]. This study is reported as per the Strengthening the Reporting of Observational Studies in Epidemiology (STROBE) guideline, specific for MR (S1 STROBE Checklist) [39].

## Results

Table 1 summarizes the characteristics of included women from UKB, ALSPAC, BiB, MoBa, and FinnGen. The SNP-insomnia associations in UKB and ALSPAC are listed in S4 Table. The 81 SNPs explained approximately 0.42% of the variance of insomnia among the 208,140 UKB women included in this study (S4 Table), and the mean F-statistic of the 81 SNPs was 11. The pooled 81 SNP-insomnia associations at 18 (OR 1.02 per effect allele, 95% CI: 1.01, 1.03, $p = 0.004$) and 32 (OR 1.02 per effect allele, 95% CI: 1.01, 1.03, $p < 0.001$) weeks of gestation in ALSPAC were in the same direction as (but weaker than) the pooled association in the original GWAS of UKB plus 23andMe women (OR 1.05 per effect allele, 95% CI: 1.05, 1.06, $p < 0.001$). The SNP-outcome associations in UKB, ALSPAC, BiB, and MoBa are listed in S5 Table.

**Table 1. Characteristics of the women in UKB, ALSPAC, BiB, MoBa, and FinnGen.**

| Variable[a] | UKB (N = 208,140) | ALSPAC (N = 6,826) | BiB (N = 2,940) | MoBa (N = 14,584) | FinnGen (N = ~123,579) |
|---|---|---|---|---|---|
| | *Mean (standard deviation)* | | | | |
| Maternal age at delivery (years) | 25.5 (4.6)[b] | 28.7(4.7) | 26.8 (6.0) | 30.0 (4.4) | Not available |
| Maternal height (cm) | 162.7 (6.2) | 164.3 (6.7) | 164.4 (6.1) | 168.3 (5.5) | Not available |
| Maternal body mass index (kg/m$^2$) | 27.0 (5.1) | 22.9 (3.7) | 26.7 (6.0) | 24.0 (4.2) | Not available |
| Gestational age (weeks) | 38.9 (3.8)[c] | 39.6 (1.7) | 39.7 (1.9) | 39.6 (1.7) | Not available |
| Offspring birthweight (grams) | 3,186.7 (547.6) | 3,441.5 (523.0) | 3,357.9 (571.2) | 3,640.8 (513.4) | Not available |
| | *N (%)* | | | | |
| Maternal education[d] | | | | | |
| O levels/GCSEs or equivalent and below | 91,093 (44.2) | 4,043 (59.5) | 1,400 (47.6) | 260 (1.9) | Not available |
| A levels/AS levels or equivalent | 48,059 (23.3) | 1,719 (25.3) | 485 (16.5) | 4,356 (31.8) | Not available |
| College or university degree | 66,873 (32.5) | 1,035 (15.2) | 551 (18.7) | 9,072 (66.3) | Not available |
| Maternal ever smoking | 85,501 (41.3) | 1,450 (21.6)[e] | 911 (31.0)[e] | 1,106 (8.8)[e] | Not available |
| Maternal ever drinking | 191,010 (91.2) | 4,580 (70.2)[e] | 1,793 (61.0)[e] | 3,644 (29.7)[e] | Not available |
| Offspring sex, male | Not available | 3,430 (50.2) | 1,504 (51.2) | 7,412 (50.9) | Not available |
| Number with fetal genotype data | 0 | 4,625 (67.8) | 1,855 (63.1) | 12,183 (83.5) | Not available |
| | N cases/N controls (Prevalence, %) | | | | |
| History of stillbirth | 4,907/139,034 (3.4) | 48/4,546 (1.0) | 31/2,588 (1.2) | 51/9,998 (0.5) | Not available |
| History of miscarriage | 42,717/139,034 (23.5) | 1,378/4,546 (23.3) | 14/2,588 (0.5) | 2,677/9,998 (21.1) | 9,113/89,340 (9.3) |
| GD | 726/200,536 (0.4) | 34/6,283 (0.5) | 136/2,657 (4.9) | 113/14,375 (0.8) | 5,687/117,892 (4.6) |
| HDP | 2,138/206,002 (1.0) | 1,099/5,698 (16.2) | 347/2,159 (13.8) | 1,892/12,652 (13.0) | 4,255/114,735 (3.6) |
| Perinatal depression | 5,178/25,130 (17.1) | 423/5,896 (6.2) | 312/2,245 (12.2) | 579/13,865 (4.0) | Not available |
| PTB | 556/4,862 (10.3) [c] | 285/4,931 (5.5) | 172/2,706 (6.0) | 495/12,846 (3.7) | 5,480/98,626 (5.3) |
| LBW | 13,429/149,084 (8.3) | 337/6,376 (5.0) | 167/2,725 (5.8) | 245/13,690 (1.8) | Not available |
| HBW | 2,716/149,084 (1.8) | 113/6,376 (1.7) | 42/2,725 (1.5) | 621/13,690 (4.3) | Not available |

[a]In UKB, these variables were measured at the recruitment that is typically 31.1 years after pregnancy.

[b]We report maternal ages at giving their first live birth. UKB women were recruited with an average age of 56.5 (standard deviation 7.9) years.

[c]Gestational age was available only in a small subset of UKB women (N = 7,280).

[d]O level, General Certificate Education (GCE) Ordinary Level; GCSE, General Certificate of Secondary Education; A level, GCE Advanced level; AS level, GCE Advanced Subsidiary level.

[e] These were maternal ever smoking/drinking in pregnancy.

ALSPAC, Avon Longitudinal Study of Parents and Children; BiB, Born in Bradford; GD, gestational diabetes; HBW, high offspring birthweight; HDP, hypertensive disorders of pregnancy; LBW, low offspring birthweight; MoBa, the Norwegian Mother, Father and Child Cohort; PTB, preterm birth; UKB, UK Biobank.

## Two-sample MR

In MR IVW combining all cohorts, point estimates for associations between lifetime susceptibility to insomnia (versus no insomnia) and outcomes ranging from ORs of 1.20 (95% CI: 0.52, 2.77, $p = 0.67$) for GD, to 3.56 (95% CI: 1.49, 8.54, $p = 0.004$) for perinatal depression (Fig 2). Despite combining data from the largest genetic studies available estimates were imprecise, with 95% CIs for all but 3 outcomes including the null. The 3 that did not include the null were miscarriage, perinatal depression, and LBW (Fig 2). S2 Fig shows IVW results for leave-one (study)-out analysis. Results were broadly consistent but dominated by large cohorts (e.g., UKB and FinnGen), with the point estimates inflated and very wide CIs in small birth cohorts. We further removed 26, 1, and 7 SNPs from analyses for stillbirth, perinatal depression, and LBW, respectively (S3 Table), because Steiger filtering suggested these SNPs potentially more associated with the respective outcome than with susceptibility to insomnia (see Methods). MR IVW estimates after Steiger filtering were consistent for perinatal depression, slightly

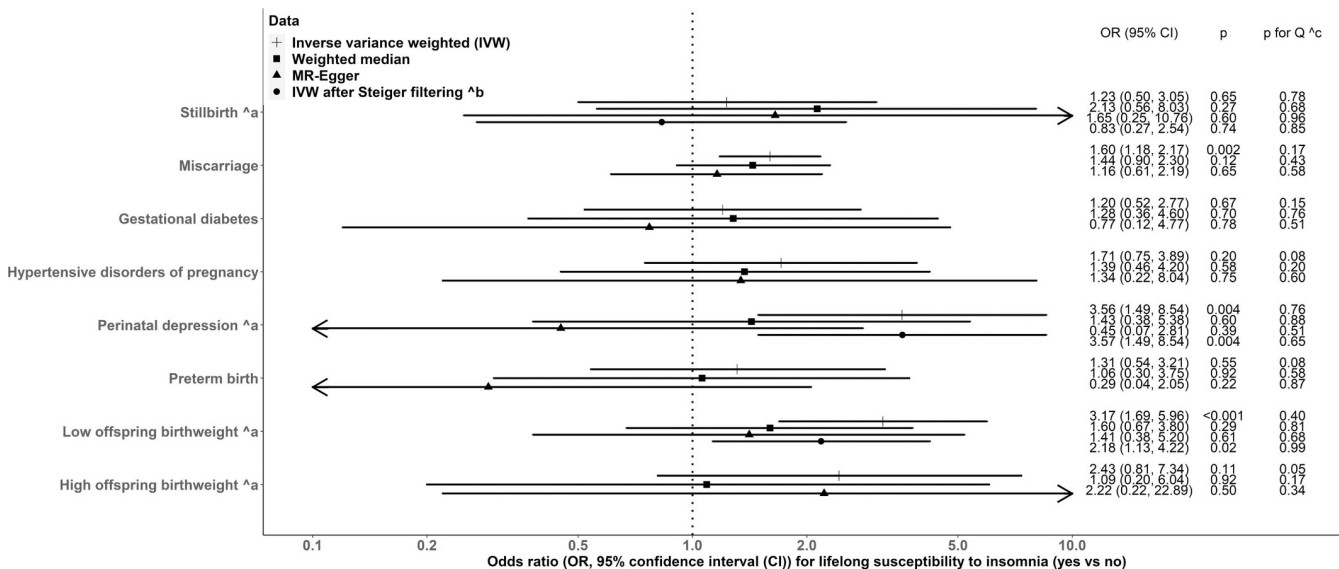

**Fig 2. Two-sample MR estimates for causal effects of insomnia on adverse pregnancy and perinatal outcomes, meta-analysing UKB, the birth cohorts, and FinnGen.** (a) For those outcomes that FinnGen did not contribute to. (b) Steiger filtering did not suggest a removal of any SNPs from MR analyses (details shown in S3 Table). (c) p for Cochran's Q-statistic <0.05 suggests between-study heterogeneity in the meta-analysis. CI, confidence interval; IVW, inverse variance weighted; MR, mendelian randomization; OR, odds ratio; SNP, single-nucleotide polymorphism; UKB, UK Biobank.

attenuated for LBW, and in the opposite direction for stillbirth with overlapped CIs both including the null (Fig 2).

Sensitivity analyses using weighted median and MR-Egger for all outcomes were direction-ally consistent with IVW but attenuated to the null for all outcomes (except stillbirth), and MR-Egger results for GD, perinatal depression and PTB were attenuated to the null (Fig 2). Between-SNP heterogeneity for MR analyses was observed with LBW and HDP (S6A Table), but leave-one (SNP)-out analyses were consistent with the main IVW estimates including all SNPs for all outcomes (S3–S5 Figs). The MR-Egger intercept *p*-value indicated unbalanced horizontal pleiotropy only for perinatal depression in UKB (S6A Table). Adjusting for fetal genotype (only possible in the birth cohorts) did not alter the SNP-outcome associations with stillbirth, miscarriage, LBW, or HBW; SNP-outcome associations with GD, HDP, and perina-tal depression were slightly attenuated; SNP-PTB associations moved slightly away from the null (S6 Fig).

After combining all cohorts, most MR estimates based on SNP-insomnia associations from linear (Fig 2) versus logistic (S6B Table) regression were in the same directions (S7 Table). An association of lifetime susceptibility to insomnia with HBW was observed using IVW (S6B Table), which previously had a wide 95% CI including the null (Fig 2).

## Multivariable regression in ALSPAC

Tables 2 and S7 summarize the characteristics of women from ALSPAC. After adjusting for maternal age, education, BMI, smoking, alcohol intake, and household occupational social class, there was an association of insomnia (versus no insomnia) at 18 weeks of gestation with perinatal depression (OR 2.96, 95% CI: 2.42, 3.63, *p* < 0.001, Fig 3). Associations with other outcomes had imprecise 95% CIs including the null, although their point estimates were in similar magnitudes to those seen in MR (Fig 3).

**Table 2. Characteristics of women in ALSPAC in confounder-adjusted multivariable regression.**

| Variable | | Insomnia at 18 weeks of gestation (N = 10,540) | |
|---|---|---|---|
| | | **Yes** | **No** |
| | | *Mean (standard deviation)* | |
| Maternal age at delivery (years) | | 27.1 (5.0) | 28.6 (4.7) |
| Maternal height (cm) | | 163.1 (6.8) | 164.3 (6.7) |
| Maternal body mass index (kg/m$^2$) | | 23.4 (4.3) | 22.9 (3.7) |
| Gestational age (weeks) | | 39.5 (1.8) | 39.6 (1.7) |
| Offspring birthweight (grams) | | 3,409.0 (546.6) | 3,446.6 (522.4) |
| | | *N (%)* | |
| Insomnia at 32 weeks of gestation | Yes | 1,073 (10.2) | 1,984 (18.8) |
| | No | 599 (5.7) | 6,781 (64.3) |
| Maternal education[a] | | | |
| O levels/GCSEs or equivalent and below | | 1,308 12.4 | 5,433 (51.5) |
| A levels/AS levels or equivalent | | 280 (2.7) | 2,126 (20.2) |
| College or university degree | | 95 (0.9) | 1,244 (11.8) |
| Household occupational social class | | | |
| I Professional occupations | | 29 (0.3) | 288 (2.7) |
| II Managerial and technical occupations | | 240 (2.3) | 2,041 (19.4) |
| III Skilled non-manual occupations | | 353 (3.3) | 2,185 (20.7) |
| III Skilled manual occupations | | 515 (4.9) | 2,444 (23.2) |
| IV Partly skilled occupations | | 321 (3.0) | 1,221 (11.6) |
| V Unskilled occupations | | 106 (1.0) | 334 (3.2) |
| Maternal smoking status in pregnancy | Ever | 590 (5.6) | 1,971 (18.7) |
| | Never | 1,106 (10.5) | 6,873 (65.2) |
| Maternal drinking status in pregnancy | Ever | 1,032 (9.8) | 5,968 (56.6) |
| | Never | 539 (5.1) | 2,556 (24.3) |
| Offspring sex | Male | 886 (8.4) | 4,539 (43.1) |
| | Female | 810 (7.7) | 4,304 (40.8) |
| GD | Case | 8 (0.1) | 36 (0.3) |
| | Control | 1,538 (14.6) | 8,210 (77.9) |
| HDP | Case | 295 (2.8) | 1,403 (13.3) |
| | Control | 1,393 (13.2) | 7,389 (70.1) |
| Perinatal depression | Case | 242 (2.3) | 460 (4.4) |
| | Control | 1,207 (11.5) | 7,759 (73.6) |
| PTB | Case | 79 (0.7) | 369 (3.5) |
| | Control | 1,170 (11.1) | 6,324 (60.0) |
| LBW | Case | 90 (0.9) | 421 (4.0) |
| | Control | 1,578 (15.0) | 8,263 (78.4) |
| HBW | Case | 28 (0.3) | 160 (1.5) |
| | Control | 1,578 (15.0) | 8,263 (78.4) |

[a]O level, General Certificate Education (GCE) Ordinary Level; GCSE, General Certificate of Secondary Education; A level, GCE Advanced level; AS level, GCE Advanced Subsidiary level.

ALSPAC, Avon Longitudinal Study of Parents and Children; GD, gestational diabetes; HBW, high offspring birthweight; HDP, hypertensive disorders of pregnancy; LBW, low offspring birthweight; PTB, preterm birth.

## Discussion

To the best of our knowledge, this is the first MR study to explore the relationship of insomnia with pregnancy and perinatal outcomes. We interpreted the MR results as reflecting a lifetime

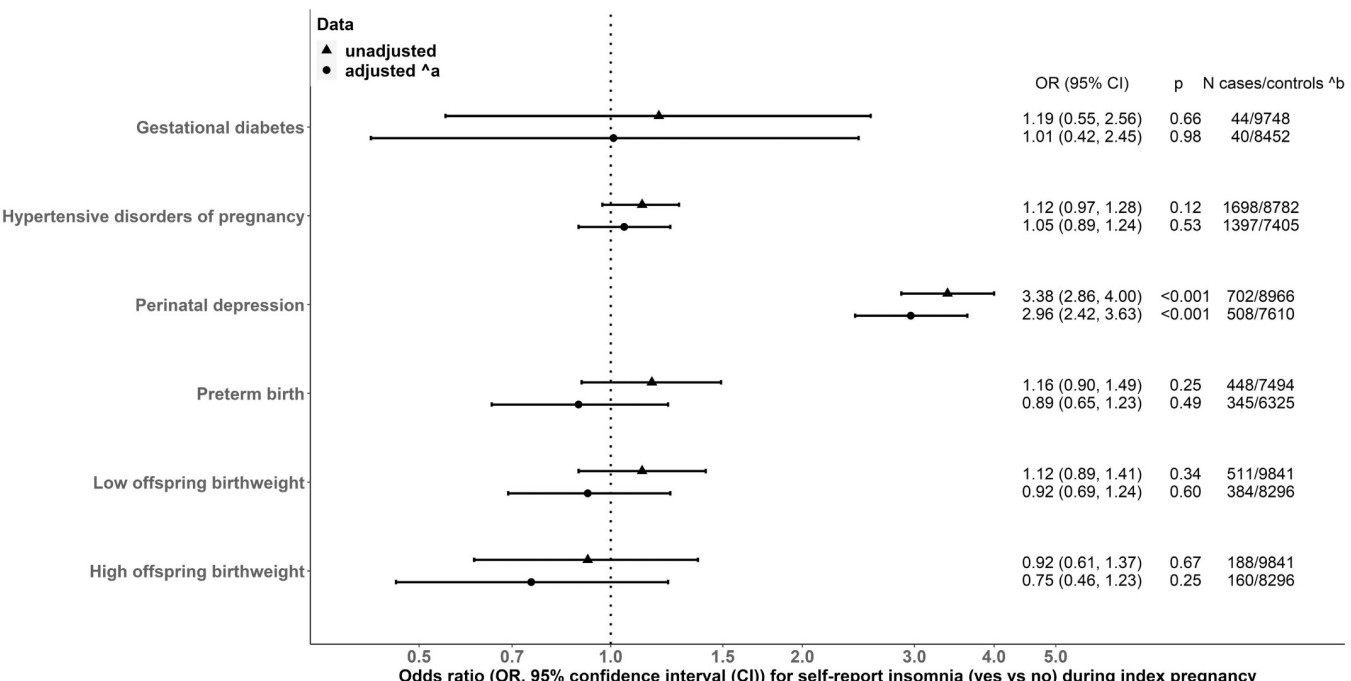

**Fig 3. Multivariable regression associations of insomnia at 18 weeks of gestation with adverse pregnancy and perinatal outcomes in ALSPAC.** (a) We adjusted for maternal age at time of delivery, education, body mass index at 12 weeks of gestation, smoking status in pregnancy and alcohol intake in the first 3 month of ALSPAC pregnancy, and household occupational social class. (b) The numbers of women in adjusted models are slightly smaller than those in crude models due to missingness (<8%) in these covariates. ALSPAC, Avon Longitudinal Study of Parents and Children; CI, confidence interval; OR, odds ratio.

susceptibility to insomnia on the basis that SNPs are determined at conception, and evidence suggested that with similar analyses of other exposures (e.g., blood pressure and C-reactive protein) this is the case [40,41]. We interpreted the multivariable regression results as reflecting associations of insomnia during pregnancy, though we could not distinguish this from pre-existing insomnia as we did not have information on sleep traits before conception. The associations of the insomnia genetic IVs with reported insomnia during pregnancy in ALSPAC provided some support that the exposures in our MR and multivariable regression analyses had some consistency with each other. Overall, our MR results provide some evidence that a lifetime susceptibility to insomnia might be associated with higher risks of miscarriage, perinatal depression, and LBW. We did not observe evidence to support associations between genetically predicted insomnia and stillbirth, GD, HDP, PTB, and HBW. In multivariable regression, we were unable to assess associations with miscarriage in the index pregnancy. Result for perinatal depression were consistent with the MR results, but this was not the case for LBW, for which no significant association with insomnia reported at 18 weeks gestation was observed.

Our findings in both MR and multivariable regression of an association of insomnia with perinatal depression are consistent with the systematic review and meta-analysis of observational studies [7], and with randomized control trials suggesting that pregnancy intervention with cognitive behavioural therapy to reduce insomnia decreases perinatal depression [8,9]. Recent systematic reviews have only identified 1 cross-sectional study of the association of insomnia with stillbirth [6,42]. This cross-sectional study compared outcomes between 190 women reporting poor sleep quality and 30 women who did not and found no association with stillbirth, although this was not the main focus of the paper [43]. We did not identify any previous studies of insomnia associations with miscarriage. Thus, our novel finding of a

possible association of insomnia with miscarriage in MR warrants replication, and larger studies that support analyses with both miscarriage and stillbirth would be valuable. Previous systematic reviews of observational associations of insomnia with GD (OR 1.37, 95% CI: 1.12, 1.69), HDP (OR 1.72, 95% CI: 1.16, 2.56), and PTB (OR 1.49, 95% CI: 1.17, 1.90) are directionally consistent but with stronger ORs than our main MR results [6]. These stronger associations could be due to insufficient adjustment of potential confounders and reverse causality, as many cross-sectional studies and unadjusted associations were included in the meta-analyses.

Several mechanisms have been suggested for why insomnia might influence pregnancy and perinatal outcomes, including insomnia resulting in increased risks of adiposity and insulin resistance that could then influence related pregnancy outcomes (GD, HDP, and HBW). Insomnia has also been suggested to influence maternal blood pressure and placental function which in turn would increase risks of HDP, miscarriage, stillbirth, and PTB. MR analyses support causal associations of insomnia with coronary heart disease, higher glycated haemoglobin, and higher glycoprotein acetyls (an inflammatory marker) in general populations of women and men [17,44,45]. Thus, an increase in cardio-metabolic risk and inflammation may mediate effects of insomnia on miscarriage and LBW, and outcomes for which our MR analyses are currently imprecise. Similarly, MR analyses have found a causal association of insomnia with depressive symptoms [17], which is coherent with our findings in relation to perinatal depression.

Key strengths of our study are that (I) to the best of our knowledge, it is the first study to use MR to explore associations of insomnia with pregnancy and perinatal outcomes; (II) we conducted confounder-adjusted multivariable regression of insomnia in pregnancy in ALSPAC—a larger sample than most previous studies; and (III) we explored a range of pregnancy and perinatal outcomes in 1 paper.

Our MR analyses may be biased by horizontal pleiotropy, particularly given our previous research showing that SNPs for insomnia are also associated with several factors that could influence pregnancy and perinatal outcomes, including education, age at first live birth, and smoking [21]. We explored this potential with a range of sensitivity analyses, including exploring between-SNP heterogeneity and using weighted median and MR-Egger methods that are more robust to such bias than IVW [33]. Results from these sensitivity analyses were broadly consistent with point estimates from IVW; however, the associations between insomnia and miscarriage, perinatal depression, and LBW no longer reached statistical significance. The wider 95% CIs observed could be attribute to the fact that those sensitivity analyses are known to have less statistical power [46]. Those attenuations towards the null could be due to weak IVs, and MR-Egger point estimates are known to be attenuated more severely than weighted median ones [13,46]. Further MR studies in larger samples with more cases would be needed for all outcomes. Specially, our results for perinatal depression require further validation using multivariable MR to account for unbalanced horizontal pleiotropy. Adjusting for fetal genotype did not alter results suggesting that bias due to fetal genotypic effects is unlikely. We did not further adjust for paternal genotype because of limited data with paternal, maternal, and offspring genotype. Furthermore, the most plausible mechanism for paternal genotype to affect pregnancy outcomes is via fetal genotype, which we have adjusted for. Interpretation of our MR estimates requires a further assumption of monotonicity in the SNP-insomnia associations. This requires that all of the women with genetic IVs related to higher susceptibility to insomnia symptoms should report more symptoms (compared to those with fewer alleles related to insomnia)—i.e., that they are "compliers" [47]. The monotonicity assumption cannot be tested. A previous study indicated potential bias when the standard IV approach (see [28]) was used for a nonlinear model [48]. In our study, using linear versus logistic regression to obtain SNP-insomnia associations showed consistent directions between MR estimates.

However, magnitudes of MR estimates cannot be compared directly due to their different scales. Further MR studies of binary exposures could apply both approaches to explore an association.

Both our MR and multivariable regression estimates could be vulnerable to selection bias, which has been extensively discussed in previous papers [25,49,50]. UKB is a selective sample (5.5% response to invitation) of adults who are healthier and better educated than the general UK adult population of the same age [51]. Information on perinatal depression and PTB was only available in a subsample of UKB women and such missingness might not be at random [52,53]. By definition our study only includes women who have experienced at least 1 pregnancy, and if insomnia influences fertility then our results might be biased [54]. However, we are not aware of robust evidence of insomnia (or SNPs related to insomnia) influencing infertility or number of children [55,56], suggesting any selection bias through only including pregnant women is unlikely to have a meaningful impact on our MR estimates [54,57].

Insomnia was measured via one self-administrated question in both UKB and ALSPAC, which could mean the binary exposure is misclassified. Non-differential misclassification of insomnia would be expected to bias MR results away from the null (given the attenuated genetic IVs-insomnia associations is the denominator), but multivariable regression results towards the null [58,59]. Similarly, there may be misclassification in some of our outcomes because of the absence of universal testing (e.g., GD in ALSPAC [60]), assessment via self-report questionnaires (e.g., birthweight in UKB), or differences between studies in definitions (e.g., in older women in UKB the gestational age thresholds for defining stillbirth and miscarriage would have differed from those used in the more contemporary birth cohorts). Non-differential misclassification of our binary outcomes would be expected to bias both MR and multivariable regression results towards the null [58,59]. Moreover, the first live-born babies of UKB women are known to be lighter than babies with various birth orders from the more contemporary birth cohorts [61,62].

Although we examined the possibility of reverse causality for individual SNPs using Steiger filtering in MR, this test could be influenced by measurement errors in insomnia and our outcomes and by confounding with opposite directions for insomnia and the outcomes [38]. Multivariable regression results for maternal outcomes could also be vulnerable to reverse causality, as tendency towards the outcomes might have influenced insomnia reported at 18 weeks of gestation. As the sources of bias in our 2 methods (MR and multivariable regression) differ, consistent results between them could strength confidence in the findings even considering different timings of an exposure [63–67]. Our previous study discussed how timings affected the interpretation of MR estimates for insomnia [21]. The similarity of the multivariable regression and MR results for GD, HDP, and perinatal depression suggests it is unlikely that residual confounding has biased regression results, horizontal pleiotropy has substantially affect MR results, or different sources of selection bias in the 2 have a strong impact, for these outcomes. The associations for PTB, LBW, and HBW were attenuated to the null compared to MR results. These suggest possible masking confounding, other biases specific to the multivariable regression, or that the MR is estimating an accumulative effect of insomnia across the life course [68], whereas the observational analyses reflect exposure only from 18 weeks of gestation to occurrence of outcome. MR analyses are statistically inefficient and despite combining relevant studies in order to increase sample size, several of our MR and multivariable regression estimates are imprecise due to small numbers of cases. Our study is limited to women of European ancestry, and we cannot assume that our results generalize to other populations.

Our findings provide some evidence for associations between insomnia and adverse pregnancy outcomes, raising the possibility that interventions to improve healthy sleep (e.g., cognitive behavioural therapy) in women of reproductive age might be beneficial to a healthy

pregnancy. However, we acknowledge the need for further MR studies based on larger GWAS of pregnancy and perinatal outcomes, larger observational studies, and studies in women from ethnic backgrounds other than white European. Further studies on the association of insomnia with recurrent miscarriage would help policy makers decide whether to allocate sleep interventions to women with a history of miscarriage when they prepare to be pregnant again.

In conclusion, our study raises the possibility of associations between insomnia and miscarriage, perinatal depression, and LBW.

## Supporting information

**S1 STROBE Checklist. STROBE-MR checklist of recommended items to address in reports of mendelian randomization studies.**
(DOCX)

**S1 Text. Descriptions of each cohort.**
(DOCX)

**S1 Fig. Flow chart of each cohort.**
(DOCX)

**S2 Fig. Leave-one (study)-out analyses of MR IVW estimates.**
(DOCX)

**S3 Fig. Leave-one-out sensitivity analysis in UK Biobank (dataset A on dataset B).**
(DOCX)

**S4 Fig. Leave-one-out sensitivity analysis in UK Biobank (dataset B on dataset A).**
(DOCX)

**S5 Fig. Leave-one-out sensitivity analysis in ALSPAC, BiB, MoBa, and FinnGen.**
(DOCX)

**S6 Fig. Maternal SNP-outcome associations comparing unadjusted to adjusted for fetal genotypes.**
(DOCX)

**S1 Table. Definitions of pregnancy and perinatal outcomes.**
(XLSX)

**S2 Table. Key characteristics of recently conducted genome-wide association studies of insomnia.**
(XLSX)

**S3 Table. Characteristics of genome-wide significant genetic variants for insomnia in women.**
(XLSX)

**S4 Table. Associations of the 81 SNPs with insomnia in UKB and ALSPAC.**
(XLSX)

**S5 Table. Associations of the 81 SNPs with outcomes in UKB, ALSPAC, BiB, and MoBa.**
(XLSX)

**S6 Table. Two-sample MR estimates in UKB and in the birth cohorts and FinnGen.**
(XLSX)

**S7 Table. Characteristics of women in ALSPAC by insomnia at 32 weeks of gestation.** (XLSX)

## Acknowledgments

This research has been conducted using the UKB Resources under application number 23938. The authors would like to thank the participants and researchers from UKB who contributed or collected data. We are extremely grateful to all the families who took part in ALSPAC, the midwives for their help in recruiting them, and the whole ALSPAC team, which includes interviewers, computer and laboratory technicians, clerical workers, research scientists, volunteers, managers, receptionists, and nurses. BiB is only possible because of the enthusiasm and commitment of the Children and Parents in BiB. We are grateful to all the participants, teachers, school staff, health professionals, and researchers who have made BiB happen. This research has been conducted using MoBa data using application number 2552. MoBa is supported by the Norwegian Ministry of Health and Care services and the Ministry of Education and Research. We are grateful to all the participating families in Norway who take part in this ongoing cohort study. We thank the Norwegian Institute of Public Health (NIPH) for generating high-quality genomic data. This research is part of the HARVEST collaboration, supported by the Research Council of Norway (#229624). We also thank the NORMENT Centre for providing genotype data, funded by the Research Council of Norway (#223273), South East Norway Health Authority, and KG Jebsen Stiftelsen. We further thank the Center for Diabetes Research, the University of Bergen for providing genotype data and performing quality control and imputation of the data funded by the ERC AdG project SELECTionPREDISPOSED, Stiftelsen Kristian Gerhard Jebsen, Trond Mohn Foundation, the Research Council of Norway, the Novo Nordisk Foundation, the University of Bergen, and the Western Norway health Authorities (Helse Vest). The authors thank FinnGen investigators for sharing their summary-level data.

## Author Contributions

**Conceptualization:** Deborah A. Lawlor.

**Data curation:** Fanny Kilpi, Ana Luiza Soares.

**Formal analysis:** Qian Yang.

**Funding acquisition:** Maria C. Magnus, Paul J. Collings, Jane West, Per Magnus, John Wright, Siri E. Håberg, Kate Tilling, Deborah A. Lawlor.

**Methodology:** Maria Carolina Borges, Eleanor Sanderson, Kate Tilling, Deborah A. Lawlor.

**Project administration:** Maria C. Magnus, Jane West, Per Magnus, John Wright, Siri E. Håberg, Deborah A. Lawlor.

**Resources:** Maria C. Magnus, Fanny Kilpi, Paul J. Collings, Ana Luiza Soares, Jane West, John Wright, Siri E. Håberg, Deborah A. Lawlor.

**Software:** Qian Yang.

**Supervision:** Maria Carolina Borges, Eleanor Sanderson, Kate Tilling, Deborah A. Lawlor.

**Visualization:** Qian Yang.

**Writing – original draft:** Qian Yang.

**Writing – review & editing:** Qian Yang, Maria Carolina Borges, Eleanor Sanderson, Maria C. Magnus, Fanny Kilpi, Paul J. Collings, Ana Luiza Soares, Jane West, Per Magnus, John Wright, Siri E. Håberg, Kate Tilling, Deborah A. Lawlor.

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
