## [Editor Report · Decision Letter 0]

9 Oct 2021

Dear Dr Yang, 

Thank you for submitting your manuscript entitled "Associations of insomnia with pregnancy and perinatal outcomes: Findings from Mendelian randomization and conventional observational studies in up to 356,069 women" for consideration by PLOS Medicine.

Your manuscript has now been evaluated by the PLOS Medicine editorial staff and I am writing to let you know that we would like to send your submission out for external peer review.

Please re-submit your manuscript within two working days, i.e. by Oct 13 2021 11:59PM.

Kind regards,

Louise Gaynor-Brook, MBBS PhD

Associate Editor

PLOS Medicine

---

## [Decision Letter · Decision Letter 1]

2 Mar 2022

Dear Dr. Yang,

Thank you very much for submitting your manuscript "Associations of insomnia with pregnancy and perinatal outcomes: Findings from Mendelian randomization and conventional observational studies in up to 356,069 women" (PMEDICINE-D-21-04244R1) for consideration at PLOS Medicine. 

Your paper was evaluated by three independent reviewers, and discussed among all the editors here and with an academic editor with relevant expertise. The reviews are appended at the bottom of this email and any accompanying reviewer attachments can be seen via the link below:

[LINK]

In light of these reviews, I am afraid that we will not be able to accept the manuscript for publication in the journal in its current form, but we would like to consider a revised version that addresses the reviewers' and editors' comments. Obviously we cannot make any decision about publication until we have seen the revised manuscript and your response, and we plan to seek re-review by one or more of the reviewers. 

We expect to receive your revised manuscript by Mar 23 2022 11:59PM. Please email us (plosmedicine@plos.org) if you have any questions or concerns.

We look forward to receiving your revised manuscript. 

Sincerely,

Louise Gaynor-Brook, MBBS PhD

PLOS Medicine

plosmedicine.org

Comments from the Academic Editor: 

In the introduction there is some conflation between insomnia and supine sleep position which I do not think is helpful. It seems likely that associations between supine sleep position and perinatal outcomes are mediated by mechanisms distinct from those described as insomnia.

In the ALPSAC logistic regression the numbers of miscarriages and stillbirths seem very high. I may have misunderstood the methodology but I think they have looked at causal effects of reported insomnia at 18 weeks and any previously reported pregnancy loss. I do not think this is correct- and would recommend that they look at the effect of insomnia reported at 18 weeks on any subsequent pregnancy complication in the same pregnancy.

In particular, I think it's necessary for the authors to fully address the concerns raised by Reviewer 2 regarding possible reverse causation, pleiotropy and shared aetiology. Some of the causal claims should also be toned down a bit.

General comments:

Please include line numbers in your revised manuscript, ideally not starting from 1 with each new page.

Your study is observational and therefore causality cannot be inferred. Please avoid use of causal language such as ‘effect’, referring to ‘associations’ instead. Please replace ‘causal effects’ with ‘causal associations’ throughout your manuscript.

Throughout the paper, please adapt reference call-outs to the following style: "... risks of gestational diabetes (GD) [5,6]..." (noting the absence of spaces within the square brackets).

Title: Please revise your title according to PLOS Medicine's style. Please place the study design in the subtitle (ie, after a colon). We suggest “Association between insomnia and pregnancy and perinatal outcomes: A Mendelian randomization analysis” or similar

Abstract Background:

Please avoid use of causal language. Please revise to “Our aim was to estimate the association between insomnia and stillbirth, miscarriage, …” 

Abstract Methods and Findings:

Please revise to “IVW showed evidence of an association between…”

Please revise to “For other outcomes IVW indicated potentially clinically important associations…”

Please provide brief demographic details of the study population (e.g. age, ethnicity, etc)

Abstract Conclusions:

Please begin your Abstract Conclusions with "In this study, we observed ..." or similar, to summarize the main findings from your study, without overstating your conclusions. Please emphasize what is new and address the implications of your study more specifically, being careful to avoid assertions of primacy. 

Please revise to “In this study, we observed evidence of causal associations…” 

Author Summary:

Please temper assertions of primacy by adding ‘to the best of our knowledge’ or similar to the third bullet point of ‘Why was this study done?’

In the final bullet point of ‘What Do These Findings Mean?’, please describe the main limitations of the study in non-technical language.

Introduction:

Please temper assertions of primacy by adding ‘to the best of our knowledge’ or similar to the penultimate paragraph. 

Please replace ‘causal effects’ with ‘causal associations’

Methods:

Did your study have a prospective protocol or analysis plan? Please state this (either way) early in the Methods section. If a prospective analysis plan (from your funding proposal, IRB or other ethics committee submission, study protocol, or other planning document written before analyzing the data) was used in designing the study, please include the relevant prospectively written document with your revised manuscript as a Supporting Information file to be published alongside your study, and cite it in the Methods section. A legend for this file should be included at the end of your manuscript. If no such document exists, please make sure that the Methods section transparently describes when analyses were planned, and if/when reported analyses differed from those that were planned. Changes in the analysis-- including those made in response to peer review comments-- should be identified as such in the Methods section of the paper, with rationale. If a reported analysis was performed based on an interesting but unanticipated pattern in the data, please be clear that the analysis was data-driven.

Please ensure that the study is reported according to the STROBE-MR guideline, and include the completed STROBE-MR checklist as Supporting Information. Please add the following statement, or similar, to the Methods: "This study is reported as per the Strengthening the Reporting of Observational Studies in Epidemiology (STROBE) guideline, specific for mendelian randomization (S1 Checklist)."

The STROBE-MR guideline can be found at https://www.strobe-mr.org/

Results: 

For the ORs presented, please specify the comparison group.

Please be clear whether the results presented in the main text are adjusted analyses and, if so, please indicate which factors are adjusted for. 

Discussion:

Please present and organize the Discussion as follows: a short, clear summary of the article's findings; what the study adds to existing research and where and why the results may differ from previous research; strengths and limitations of the study; implications and next steps for research, clinical practice, and/or public policy; one-paragraph conclusion.

Please remove all subheadings within your Discussion e.g. Study strengths and limitations

Please avoid causal language such as “effect” and “increased risk”. For example, please revise to “the strongest association using MR evidence was observed between insomnia and the risk of miscarriage, …” or similar. 

Please temper assertions of primacy by adding ‘to the best of our knowledge’ or similar to “our novel finding” and “it is the first study to use MR…”

Figures:

Please consider avoiding the use of red and green in order to make your figure more accessible to those with colour blindness.

Tables:

When a p value is given, please specify the statistical test used to determine it in the table legend.

References:

Where website addresses are cited, please specify the date of access. 

Comments from the reviewers:

Reviewer #1: This study examined the effects of insomnia on pregnancy and perinatal outcomes using two-sample Mendelian randomization design. It is a well-designed and carefully conducted study. Here are my specific comments on this study. 

1. Please provide line number for easier reference. 

2. There is competing interest that needs to be carefully reviewed

3. The data are not fully available to the public.

4. page 5, Introduction: "three RCT assessing the effects… or perinatal outcomes". I feel that this part can be briefer. No need to show the numbers here. Or you can discuss the results in more details in the discussion section.

5. Page 8, outcome measures: "if multiple pregnancies were enrolled in the birth cohorts, we randomly selected one pregnancy per woman". First, how many multiple pregnancies are there in the dataset? Second, is this the best way to deal with multiple pregnancies? Can you adjust for the multiple pregnancies using techniques such as random effect model? Also, if you have to choose one of the pregnancies, is that better to choose the first pregnancy rather than randomly select one pregnancy. I assume most the single pregnancies are the first pregnancy?

6. Page 9, insomnia measures: The definitions of insomnia in ALSPAC and in other studies are different. I would say lifelong exposure to insomnia and insomnia during pregnancy are drastically different. Is that possible to adjust for the two measurements to make them more consistent with each other? For example, using studies collecting data for both definitions to examine the relationship between them and adjust the data of two definitions accordingly. Given the difference between the two definitions, I would say that comparison between the MR and conventional approach become less useful. 

7. Page 9, SNP selection and SNP-insomnia associations: fitting linear regression on a binary outcome (i.e., insomnia) is inappropriate. Can you fit logistic or probit model and predict the log-odds of insomnia instead? I don't see why you have to worry about non-collapsibility and interpretation here. This are intermediate estimates anyway. 

8. Page 13: "if the MR-Egger intercept is not null it suggests the presence of horizontal pleiotropy, and the MR-Egger slope provides an effect estimate corrected for unbalanced horizontal pleiotropy" Then, why not use the MR-Egger directly to avoid making assumption on horizontal pleiotropy? Given the sample size you have, do you really have to worry about statistical power here? 

Reviewer #2: The paper applies Mendelian randomization and multivariable linear regression to study the effects of lifetime tendency for insomnia on pregnancy related outcomes. Overall, I think this is a well written and presented study, and an interesting use of Mendelian randomization methods. There is some evidence of an effect of insomnia on some pregnancy outcomes, although many of the methods used are reasonably low powered and sensitivity analyses do not always support the primary analyses. Nonetheless, the authors place the results in the context of existing literature and their results provide the basis for future study of the identified associations.

I have the following comments.

- Although it is noted in the Discussion, I feel that potential reverse causation could be particularly relevant here, especially given that in many cases it is unknown if the exposure occurred before or after the outcome. A bidirectional analysis and/or the MR Steiger test of directionality could be used to analyse potential reverse causation. 

- The Discussion suggests there are known risk factors which may sit on pleiotropic pathways from the genetic instruments to the outcomes, and hence be causing bias in the MR results. Do the genetic instruments associate with any of these possibly pleiotropic risk factors? The conclusion regarding the effect of insomnia on perinatal depression seems somewhat weak: there is evidence of unbalanced pleiotropy from the MR-Egger intercept test, and the direction of the MR-Egger point estimate is inconsistent with the MR-IVW result. Could these results be explained by horizontal pleiotropy, and if so, could this be explored in a multivariable MR analysis?

- Three SNPs were excluded from the BiB dataset due to having a minor allele frequency < 0.01. Does this indicate that the genetic distribution of individuals in BiB is not similar to the other cohorts? For example, rs10947428 has a minor allele frequency of approx 0.2 in the other datasets, which seems to be a big difference to BiB. Is it therefore valid to combine BiB with the other datasets? Also, rs9943753 is given in the text as a SNP which is excluded from BiB, but doesn't seem to be in the list of instruments in the supplementary tables.

- For the leave-one-out analysis of cohorts shown in S2 Fig, it might be relevant to also do this analysis without UKB. UKB dominates the others (due to its size), so I don't think the analysis is able to show whether there is heterogeneity between the birth cohorts. This may, for example, help to show whether BiB is from a similar population group to the other cohorts or not.

- Page 13: "if the MR-Egger intercept is not null it suggests the presence of horizontal pleiotropy" - should "horizontal" be "unbalanced" in this sentence?

Reviewer #3: Thank you for the opportunity to review this manuscript. The paper addresses an important question, whether sleep disturbances during pregnancy is causally linked to adverse pregnancy outcomes. Insomnia is becoming increasingly common, and is recognized to be associated with a variety of adverse health outcomes. As highlighted by the authors, several epidemiological studies have highlighted potential links between sleep disorders including insomnia and pregnancy outcome, though whether insomnia is causally linked to these outcomes are not clear. 

The authors sought to address this series of research question using several datasets, including UK Biobank, 3 birth cohorts (ALSPAC, BIB, MoBa) and FinnGen. Results obtained from the different datasets were then meta-analyzed, as appropriate. They also performed multivariable regression on association between insomnia, as reported at 18 weeks gestation and 32 weeks gestation, with different pregnancy outcome, with adjustment of the known confounders. The main findings were significant, and clinically relevant association between insomnia and miscarriage, perinatal depression, and low birthweight. Some of the sensitivity analyses gave additional support, though some of the other analyses had CI that overlapped with null.

The study is overall well-performed, and utilized several of the largest datasets available to address the research question. The outcome definitions have been harmonized. The MR methodology and sensitivity analyses are clearly described, as would be expected given the extensive expertise on MR from this group of investigators from Bristol. 

Results from the main analysis (using IVW), and the sensitivity analyses using weighted median and MR-Egger, were mostly consistent in direction and size of effects. Perhaps rather surprisingly, the confidence interval for the insomnia-adverse pregnancy outcome and perinatal outcome analysis was quite wide for the majority of outcomes, despite the large sample size. The between-study heterogeneity appeared limited for most outcomes, suggesting the source of the variation is prob not due to study differences. 

Some limitations include the timing of the assessment of insomnia, which was at 18 and 32weeks gestation in ALSPAC, but with history of stillbirth or history of miscarriage used as the outcome. Another major limitation is that insomnia was defined according to self-reported measures.

Major comments

1. The confidence intervals for the estimates were rather large for most outcomes examined despite the large sample size. Can the authors discuss and speculate more on why this would be the case?

2. In general, genetically determined exposures (e.g. insomnia in this case) would show larger effects by the instrumental variable (reflecting lifelong predisposition to the exposure) compared to effect size observed in the multivariable regression analysis. This was however not the case for the outcomes examined in this analysis. Is this related to the impact of the exposure being limited to that during pregnancy. Perhaps some brief discussion of this would be useful.

[LINK]

---

## [Decision Letter · Decision Letter 2]

26 May 2022

Dear Dr. Yang,

Thank you very much for re-submitting your manuscript "Associations between insomnia and pregnancy and perinatal outcomes:  Mendelian randomization and conventional observational analyses" (PMEDICINE-D-21-04244R2) for consideration at PLOS Medicine. 

Your revised paper was evaluated by a senior editor and discussed among all the editors here. It was also discussed with an academic editor with relevant expertise, and sent to two of the original reviewers, including a statistical reviewer. The reviews are appended at the bottom of this email and any accompanying reviewer attachments can be seen via the link below:

[LINK]

In light of the remaining comments from Reviewer 1, I am afraid that we will not be able to accept the manuscript for publication in the journal in its current form, but we would like to consider a revised version that addresses the reviewers' and editors' comments. Obviously we cannot make any decision about publication until we have seen the revised manuscript and your response, and we plan to seek re-review by one or more of the reviewers. 

We expect to receive your revised manuscript by Jun 02 2022 11:59PM. Please email us (plosmedicine@plos.org) if you have any questions or concerns.

We look forward to receiving your revised manuscript. 

Sincerely,

Caitlin Moyer, PhD

Associate Editor

PLOS Medicine

on behalf of,

Louise Gaynor-Brook, MBBS PhD

PLOS Medicine

plosmedicine.org

Points for a second major:

1. Response to reviewers: Please completely address the remaining points of reviewer 1.

2. Title: We suggest: “Associations between insomnia and pregnancy and perinatal outcomes: Evidence from Mendelian randomization and observational analyses” or similar.

3. Abstract: Background: If feasible, it might be helpful to add a few words or a sentences to explain the gap in knowledge being addressed by your study, particularly the MR aspect (given that you point out previous work has already revealed observational evidence for associations between insomnia and adverse pregnancy/perinatal outcomes).

4. Abstract: Methods and Findings: Please explicitly mention the main outcome measures.

5. Abstract: Methods and Findings: Please quantify the main results (with 95% CIs and p values).

6. Abstract: Line 33: Please define LBW at first use.

7. Abstract: Methods and Findings: Line 33-34: In light of the results, please revise the wording to: “For other outcomes IVW results did not support associations with insomnia, yielding observations that were directionally similar (OR range 1.20 to 2.43), but with 95% CIs that were wide and included the null.” or similar. It may also help to mention which specific outcomes are being described here, and please report OR, 95% CI, and p values for each.

8. Abstract: Line 35: We suggest revising to: “Associations between genetically predicted insomnia and miscarriage, perinatal depression, and LBW were not observed in weighted median and MR Egger analyses. Results from these sensitivity analyses were directionally consistent with IVW analyses for all outcomes, with the exception of gestational diabetes, perinatal depression, and preterm birth.

9. Abstract: Methods and Findings: Would it be considered a limitation also that genetic instruments for insomnia were not specific to women in pregnancy (could there be different mechanisms related to pregnancy insomnia)?

10. Abstract: Conclusions: Line 42: Please revise the wording to “...we observed some evidence supporting a possible causal relationship between genetically-predicted insomnia and miscarriage, perinatal depression, and LBW.” We suggest also mentioning that you also observed evidence supporting a relationship between insomnia at 18 weeks gestation and perinatal depression, but no observational evidence to support other outcomes.

11. Abstract: Conclusions: Here and throughout (also Line 67-68), please rephrase, and avoid wording that implies “insomnia causes miscarriage” or similar.

12. Author summary: Line 63: Please remove the word “consistent” given the findings from the various sensitivity analyses.

13. Author summary: Line 64-65: We suggest re-wording to: “...and MR analyses also suggested evidence for an association between genetically-predicted insomnia and risk of miscarriage and low offspring birthweight.”

14. Author summary: Line 67: We suggest: “These findings raise the possibility that interventions to improve healthy sleep may be beneficial for a healthy pregnancy.”

15. Introduction: Line 89-90: Please clarify this sentence: “As disturbed sleep is a symptom of depression it is unclear whether these studies reflect a causal effect of insomnia, or it is part of the diagnostic criteria.”

16. Methods: Line 130-133: Thank you for mentioning the lack of a pre-specified analysis plan. Please also make sure that the Methods section transparently describes when analyses were planned, and when/why any data-driven changes to analyses took place. 

17. Results: Throughout this section, please present the OR, with 95% CIs and p values, for all main outcomes described.

18. Results: Line 309: Please mention here that results of IVW analyses were attenuated for all associations identified in weighted median and MR-Egger analyses.

19. Results: Lines 319-325: Please present the OR, 95% CIs, p values for associations with the primary outcomes in the text for the multivariable regression analyses.

20. Results: Lines 314-317: Please describe in the text, or in the S6 Fig legend, how it was established whether adjustment for fetal genotype did or did not significantly alter SNP-outcome associations.

21. Discussion: Line 340-348: Given the potential for this to be misinterpreted, we suggest rewording to clarify: “Overall, our MR results provide evidence that a lifetime susceptibility to insomnia might associate with higher risks of miscarriage, perintatal depression, and LBW. We did not observe evidence to support associations between genetically-predicted insomnia and stillbirth, GD, HDP, PTB, and HBW. In multivariable regression, we were unable to assess associations with miscarriage in the index pregnancy. Results for perinatal depression were consistent with the MR results, but this was not the case for LBW, for which no significant association with insomnia reported at 18 weeks gestation was observed.”

22. Discussion: Line 352-353: “Whilst previous studies have shown that pregnancy supine sleeping position (which is associated with insomnia) to be associated with stillbirth [6,39]” We suggest removing this, as per the discussion in the rebuttal letter.

23. Discussion: Line 387-388: “Results from these sensitivity analyses were broadly consistent with our main

388 analyses but were less precise (i.e. had wider CIs).” Please provide a sentence or so of discussion on the implications of the fact that the results from the IVW analysis were not found to be robust in sensitivity analyses, in terms of how that affects your interpretations.

24. Discussion: Please present and organize the Discussion as follows: a short, clear summary of the article's findings; what the study adds to existing research and where and why the results may differ from previous research; strengths and limitations of the study; implications and next steps for research, clinical practice, and/or public policy; one-paragraph conclusion. Specifically, please include a paragraph discussing the implications, next steps for research, clinical practice, and/or public policy between the discussion of study limitations and the overall conclusion paragraph.

25. Discussion: Line 438: We suggest revising to: “In conclusion, our study raises the possibility of associations between insomnia and miscarriage, perinatal depression and LBW.”

26. Line 443: Data availability statement: Please remove this section from the main text of the manuscript, and make sure all information is completely and accurately entered into the data availability section of the manuscript submission system.

27. Line 460: Funding: Please remove this section from the main text of the manuscript, and make sure all information is completely and accurately entered into the funding section of the manuscript submission system.

28. Line 486: Competing interests: Please remove this section from the main text of the manuscript, and make sure all information is completely and accurately entered into the competing interests section of the manuscript submission system.

29. References: Please use the "Vancouver" style for reference formatting, and see our website for other reference guidelines https://journals.plos.org/plosmedicine/s/submission-guidelines#loc-references

For reference 23, please provide complete information for the reference. 

For reference 36, the journal abbreviation should be BMJ. 

For reference 37, the journal abbreviation should be JAMA. 

For reference 57, the journal abbreviation should be BJOG. 

For reference 64, the journal abbreviation should be Nat Rev Methods Primers.

30. Table 1: It might be helpful to clarify in the legend that “miscarriage/stillbirth” represent any history of those (if this is accurate).

31. Figure 2: Please present the results taking Steiger filtering into account for all of the outcomes, not just those that were significant. Please present this in the figure rather than mentioning the results in the legend. Please present p values in addition to 95% CIs for each association.

32. Figure 3: Please correct the title to: “...insomnia at 18 weeks of gestation…”Please present p values in addition to 95% CIs for each association.

33. S3 Table: Please clarify where the footnotes indicating the Steiger filtering of individual SNPs are indicated (e.g. footnotes d, e, f were not found in the table).

34. S7 Table: Please move this table to the main text (at least the 18 weeks gestation dataset). Please clarify what is meant by “Insomnia at the other time point” in the variables list.

35. S6 Fig. Please display all plots on the same axis scale. Please describe interpretations in the figure legend (e.g. attenuation of relationships).

Comments from the reviewers:

Reviewer #1: Thank you for the thoughtful responses from the authors. I think the authors have addressed most concerns from me and other reviewers. However, there are still a few remaining concerns I have about the current manuscript. 

1. Response to my point # 5: 

The explanations given by the authors are great. I think it would be great to include these detailed explanations in the manuscript (possibly in the discussion section). The current revisions made by the authors are insufficient to me. 

2. Response to my point # 6: 

(page 27, line 474) We added: "As the sources of bias in our two methods (MR and multivariable regression) differ, where we have consistency between them this increases our confidence in those consistent results being the causal association [17-21]."

I would still be cautious about making any causal inference/claim here based on the consistency in the results.

3. Response to my point # 7: 

Using conventional linear IV method in nonlinear models can lead to substantial bias (https://pubmed.ncbi.nlm.nih.gov/18546544/). Also, as I mentioned, since estimates of the association between SNP and insomnia are just intermediate results for the IV approach, why do we care so much about the interpretation of the intermediate results? Should we prioritize having unbiased estimates of association between insomnia and perinatal outcomes as these associations are more important? 

Reviewer #2: I thank the authors for their response to my comments, which have all been addressed satisfactorily.

[LINK]

---

## [Decision Letter · Decision Letter 3]

4 Aug 2022

Dear Dr. Yang,

Thank you very much for re-submitting your manuscript "Associations between insomnia and pregnancy and perinatal outcomes:  Evidence from Mendelian randomization and multivariable regression analyses" (PMEDICINE-D-21-04244R3) for review by PLOS Medicine.

I have discussed the paper with my colleagues and the academic editor and it was also seen again by one of the reviewers. I am pleased to say that provided the remaining editorial and production issues are dealt with we are planning to accept the paper for publication in the journal.

[LINK]

We look forward to receiving the revised manuscript by Aug 11 2022 11:59PM.   

Sincerely,

Caitlin Moyer, PhD

Associate Editor 

PLOS Medicine

plosmedicine.org

Requests from Editors:

1. Abstract: Methods and Findings: Please explicitly state the outcomes of interest early in this section when describing the study methods: ever experiencing stillbirth, ever experiencing miscarriage, GD, HDP, perinatal depression, PTB (gestational age <37 completed weeks), LBW (<2,500 grams) and HBW (>4,500 grams).

2. Abstract: Methods and Findings: Line 34, Line 41: Here, and throughout, please report p values as p<0.001 where applicable, unless there is a statistical reason to present exact p values smaller than this.

3. Abstract: Line 35: Please define “HDP” at first use in the text.

4. Abstract: Conclusions: Line 46-47: Please revise to: “In this study, we observed evidence s

5. Author summary: Line 59-60: Please revise to: “To the best of our knowledge, Mendelian randomization (MR) has not been used to explore whether there is evidence to support a causal association between insomnia and adverse pregnancy and perinatal outcomes.”

6. Author summary: We suggest starting off this point to focus on the interpretation of the study findings: “These findings raise the possibility that insomnia may be related to pregnancy outcomes…”.

7. Results: Line 294 and 296: Here, and throughout the Results section, please use p<0.001 where appropriate.

8. Results: Line 299-301: Please revise to: “In MR IVW combining all cohorts, point estimates for associations between lifetime susceptibility to insomnia (vs no insomnia) and outcomes ranged from ORs of 1.20 (95% CI: 0.52, 2.77, p value=0.67) for GD, to 3.56 (95% CI: 1.49, 8.54, p-value=0.004) for perinatal depression (Fig 2).”

9. Results: Line 309: Please revise to: “MR IVW estimates after Steiger filtering were consistent for…”

10. Discussion: Line 362: Please revise to “...provide some evidence that a lifetime susceptibility to insomnia might be associated with….”

11. Discussion: Line 405-406: Please revise to: “Results from these sensitivity analyses were broadly consistent with point estimates from IVW, however, the associations between insomnia and miscarriage, perinatal depression, and low offspring birth weight no longer reached statistical significance. The wider 95% CIs observed could be attributable to the fact that these sensitivity analyses are known to be statistically less efficient [45].” It might be helpful to provide a sentence of explanation of what is meant by statistically less efficient, and alternative interpretations.

12. Discussion: Line 464: We suggest revising the first sentence to more closely address the implications of the study without overreaching what can be concluded (i.e. the study does not exactly provide evidence that suggests a sleep intervention could be beneficial in healthy pregnancy). We suggest: “Our findings provide some evidence for associations between insomnia and pregnancy outcomes, raising the possibility that interventions to improve healthy sleep (e.g. cognitive behavioural therapy) in women of reproductive age might be beneficial strategy to promote healthy pregnancy.” or similar.

13. References: When providing a DOI please include both the label and full DOI included at the end of the reference (e.g. doi: 10.1016/j.molimm.2014.11.005). Please do not provide a shortened DOI or the URL.

14. Figure 3: Please use p<0.001 for p values when appropriate (for perinatal depression).

15. S6 Table: Please provide p values for associations reported.

16. Table 1, Table 2, and S7 Table: Please define the abbreviations used for maternal education (GCSE) in the legend.

17. S2 Text: We suggest moving this to the main text of the Methods section at line 206.

Comments from Reviewers:

Reviewer #1: Thank you for the response. I think the authors have adequately addressed my comments from last round. I would recommend acceptance of the paper. Congratulations!

[LINK]

---

## [Editor Report · Decision Letter 4]

15 Aug 2022

Dear Dr Yang, 

On behalf of my colleagues and the Academic Editor, Sarah Stock, I am pleased to inform you that we have agreed to publish your manuscript "Associations between insomnia and pregnancy and perinatal outcomes:  Evidence from Mendelian randomization and multivariable regression analyses" (PMEDICINE-D-21-04244R4) in PLOS Medicine.

Please address the following editorial points:

-Abstract: Line 48-49: Apologies for the confusion with the previous comment on this sentence. We suggest revising to: “In this study, we observed some evidence in support of a possible causal relationship between genetically-predicted insomnia and miscarriage, perinatal depression, and LBW. Our study also found observational evidence in support of an association between insomnia in pregnancy and perinatal depression, with no clear multivariable evidence of an association with LBW.”

-Abstract: Line 52: Please remove the word “improving” from the sentence.

-Author summary: Line 59: Please change to “not clear” in this sentence.

-Supporting information files: Please remove the unpublished manuscript from the supporting information files.

PRESS

Sincerely, 

Caitlin Moyer, Ph.D. 

Associate Editor 

PLOS Medicine